# The Prevalence of Smoking, Determinants and Chance of Psychological Problems among Smokers in an Urban Community Housing Project in Malaysia

**DOI:** 10.3390/ijerph16101762

**Published:** 2019-05-18

**Authors:** Rusdi Abd Rashid, Sharmilla Kanagasundram, Mahmoud Danaee, Hazreen Abdul Majid, Ahmad Hatim Sulaiman, Muhammad Muhsin Ahmad Zahari, Chong Guan Ng, Benedict Francis, Wan Azlinda Irnee Wan Husin, Tin Tin Su

**Affiliations:** 1University of Malaya Centre of Addiction Sciences (UMCAS), University of Malaya, Kuala Lumpur 50603, Malaysia; rusdi@ummc.edu.my (R.A.R.); irnee84_umcas@um.edu.my (W.A.I.W.H.); 2Department of Psychological Medicine, Faculty of Medicine, University of Malaya, Kuala Lumpur 50603, Malaysia; hatim@um.edu.my (A.H.S.); muhsin@ummc.edu.my (M.M.A.Z.); chong_guan@um.edu.my (C.G.N.); ben.franciscan@gmail.com (B.F.); 3Department of Social and Preventive Medicine, Faculty of Medicine, University of Malaya, Kuala Lumpur 50603, Malaysia; mdanaee@um.edu.my (M.D.); hazreen@ummc.edu.my (H.A.M.); 4Centre for Population Health (CePH), Department of Social and Preventive Medicine, Faculty of Medicine, University of Malaya, Kuala Lumpur 50603, Malaysia; tstin@ummc.edu.my; 5South East Asia Community Observatory (SEACO), Monash University Malaysia, Bandar Sunway 47500, Malaysia

**Keywords:** smoking, self-medication, depressive behaviors, poor coping

## Abstract

*Objective:* This study was conducted to assess the prevalence, pattern of smoking and sociodemographic factors among Kerinchi residents in Kuala Lumpur, as well as to identify the association between smoking, stress, anxiety and depression. *Methods:* This study was carried out at four community housing projects in the Lembah Pantai area in Kuala Lumpur. Data was collected between 3 February 2012, and 29 November 2012. Data collectors made house visits and used interviewer administered questionnaires containing questions on demographic data and smoking patterns. Depression anxiety stress scale (DASS) was used to assess psychological symptoms. Alcohol smoking and substance involvement screening tool (ASSIST) scale was used to assess nicotine use. *Results:* Data from 1989 individuals (833 households) showed the age of respondents ranged from 18 to 89 years and the mean age was 39.12 years. There were 316 smokers indicating the prevalence of smoking was 15.85%, with 35.5% among males and 1.8% among females. Further, 86.6% of smokers were Malay and 87% were Muslims. Divorce was associated with smoking. Unemployment and housewives were less associated with smoking. Depression and anxiety were significantly associated with smoking (OR = 1.347. 95% CI: 1.042–1.741) and (OR = 1.401. 95% CI: 1.095–1.793) respectively. *Conclusion:* Screening for depression and anxiety should be routinely performed in the primary care setting and in population-based health screening to intervene early in patients who smoke.

## 1. Introduction

According to the literature, tobacco use started in the fifteenth century. Initially, it was used for pleasure and was even proclaimed to have medicinal uses [1]. However, it is now known that there are many hazardous compounds in cigarette smoke [2] which cause a great deal of morbidity [3]. It is of great concern that those individuals who are unable to cope with mental stress use tobacco to get psychological relief [4]. This is due to the fact that tobacco has been shown to have antidepressant effects [5]. However, this is a maladaptive way of coping as smoking has been shown to decrease life span [6]. Fortunately, statistics show that cigarette use in Malaysia has been decreasing over the last few years [7].

Tobacco use is one of the major preventable areas of morbidity and this lead to considerable research being undertaken to assess the prevalence of its use in Malaysia and the world [8]. A study by Hock et al. (2013) [7] showed that 46.5% of the adult population in Malaysia aged 18 years and above smoked. Social determinants have been shown to play a large role. Smoking is not equal among the various social determinants, and is prevalent among people living in poverty [9].

Previous research has shown that nicotine has strong associations with psychiatric morbidity such as stress, anxiety and depression [10,11,12]. People living in poverty have less access to healthcare [13] and hence, nicotine becomes a useful way for them to self- medicate. This alarming fact brings an urgency for health care providers and researchers to find specific interventions and comprehensive health programs in order to overcome this issue. In order to overcome this extensive problem, specific knowledge such as demographic details and targeted intervention, is required to prevent or mitigate the usage of nicotine.

Research indicates that 80% of smokers come from low and middle income countries [9]. The Kerinchi community was chosen because Kerinchi is highly populated and comprises people of lower socioeconomic status compared to other areas in Kuala Lumpur. A low level of education and low income, makes one susceptible to smoking [14,15]. In addition to the above mentioned factors, other demographic characteristics such as age, gender, marital status, religion and size of the family are factors that may influence the use of the substance [7,16]. Thus, the importance of this study is to shed light on the number, characteristics and pattern of smokers among the Kerinchi community, and also assess if there is any association between smoking and stress, anxiety and depression.

### 1.1. Objective


(1)To identify the prevalence and pattern of smoking among Kerinchi residents in Kuala Lumpur.(2)To identify the association between sociodemographic factors and smoking among Kerinchi residents in Kuala Lumpur.(3)To identify the association between stress, anxiety and depression, and smoking among Kerinchi residents in Kuala Lumpur.


### 1.2. Hypothesis

There is an association between prevalence of smoking, socioeconomic factors, depression and anxiety among smokers residing in Kerinchi, Kuala Lumpur.

## 2. Methodology

### 2.1. Setting

This study was carried out at 4 community housing projects, that is, Projek Perumahan Rakyat (PPR) in the Lembah Pantai area in Kuala Lumpur were the population sample in this study. There were four PPR involved which were PPR Kerinchi, PPR Seri Cempaka, PPR Pantai Ria and PPR Seri Pantai. This study was funded by the Municipality in Kuala Lumpur as part of the squatter resettlement program. The requirements to obtain a housing unit in the PPR were that a family (1) must have at least 1 child, (2) should earn less that RM 2000 (RM = Ringgit Malaysia) and (3) should not own a property within 35 km from the capital city. All data collection procedures were approved by the Institutional Review Board at University of Malaya (IRB protocol MEC Ref number 890.161).

### 2.2. Data Collection

The sampling method used in this study was universal sampling and therefore all participants from the households surveyed were included in this study. The time frame for data collection was 3 February 2012, to 29 November 2012. Data collectors went from house to house to collect data using interviewer administered questionnaires. The questionnaires which were in Malay and English contained questions on demographic data and smoking patterns. In addition, Depression anxiety stress scale (DASS) was used to assess psychological symptoms in the sample. Alcohol smoking and substance involvement screening tool (ASSIST) was used to assess nicotine use.

Psychological problems: Psychological problems were characterized as being stress, depression and anxiety [17].

Nicotine Abuse: Maladaptive pattern of use indicated by continued use despite knowledge of having a persistent or recurrent social, occupational, psychological or physical problem that is caused or exacerbated by the recurrent use in situations in which it is physically hazardous [18].

Poverty: People that have less information, less resource, less able to access health care, less nourished, have a higher risk of illness and disability (World Health Organization, 2013) [19].

### 2.3. Scales Used

Depression anxiety stress scales (DASS) consist of a list of 21 symptoms, each of which is to be rated on a four point scale of how much the patient had that symptom in the last week [20]. This scale is investigator administered.

Alcohol smoking and substance involvement screening tool (ASSIST) scale was used to assess nicotine abuse [21].

### 2.4. Statistical Analyses

All statistical analyses were performed using IBM SPSS Statistics for Windows, version 25.0 (IBM Corp LP, Armonk, NY, USA). Descriptive statistics, such as frequency and percentage, were applied for categorical data. Inferential statistics, including Chi square test, univariate and stepwise multivariate logistic regression, were carried out to study the relationship between variables and outcome (smoker vs non-smoker). A *p*-value of ≤ 0.05 was considered statistically significant.

## 3. Results

### 3.1. Descriptive Analyses

#### 3.1.1. To Identify the Prevalence and Pattern of Smoking among Kerinchi Residents in Kuala Lumpur

The data from 1989 individuals (833 households) were collected in the household survey in the PRPs. The age of respondents in this study ranged from 18 years to 89 years. The mean age of the respondents in this study was 39.12. Descriptive analysis was completed to analyze the demographic characteristic of the respondents (Table 1). Slightly more than half of the respondents in this study were female, that is, 1054 (53.0%) were female and 935 (47.0%) were male. In respect of ethnicity, most of the respondents were Malay, with 1608 (80.8%) being Malay, followed by 339 (17%) being Indian and 21 (1.1%) being Chinese.

A total of 351 respondents were tobacco product users and 316 were smokers which indicated that the prevalence of smoking among Kerinchi residents in Kuala Lumpur was 15.85%, and tobacco product use was at 17.64%. Prevalence of smoking among males was 35.5% and only 1.8% of females smoked. Among smokers, 86.6% were Malay, 87% were Muslims, 84% were male and 29% were in the 18 to 29 age group. Regarding the other Bumiputra, 2 were from Sabah and 3 from Sarawak.

Several characteristics related to smoking behaviors were studied (Table 2) and results showed that the majority of respondents were introduced initially through friends (75.8%). Regarding the frequency of usage, 82.3% smoked daily.

#### 3.1.2. To Identify the Association between Sociodemographic Factors and Smoking among Kerinchi Residents in Kuala Lumpur

To evaluate the relationship between sociodemographic variables and smoking status (Table 3), Chi square test was used. The results indicated that, except for age (*χ^2^* = 3.851, *p* = 0.427) and educational level (*χ^2^* = 6.959, *p* = 0.037) which were not significantly correlated with smoking status, all other variables were significantly associated with smoking status. Regarding gender, the majority of smokers were male (94.6%) and results indicated that there is a significant association between gender and smoking (*χ^2^* = 387.315, *p* < 0.001). According to these results, it was found that smoking status was significantly correlated with ethnicity (*χ^2^* = 10.192, *p* = 0.037), religion (*χ^2^* = 9.935, *p* = 0.042), marital status (*χ^2^* = 31.845, *p* < 0.001) and occupational status (*χ^2^* = 120.961, *p* < 0.001). 

Based on the results of univariate analysis, significantly associated factors were applied in a stepwise multivariate logistic regression and the results showed that four factors significantly predicted smoking among respondents. According to these results, females had a significantly less chance of being a smoker compared to a male respondent (OR = 0.39. 95% CI: 0.023–0.066). Ethnic groups were not significantly affected by smoking. The chance of smoking among divorced respondents was significantly higher than married respondents (OR = 1.999. 95% CI: 1.310–3.050). The last significant factor was occupation in which only two groups, housewives (OR = 0.242. 95% CI: 0.097–0.606) and those unemployed (OR = 0.228. 95% CI: 0.072–0.723) had a lower chance of smoking compared to a civil servant as a reference group.

#### 3.1.3. To Identify the Association between Depression Anxiety and Stress Scales and Smoking among Kerinchi Residents in Kuala Lumpur

The last research objective for the current study was related to investigating the relationship between depression, anxiety and stress with smoking. Results of the multivariate logistic regression (Table 4) showed that respondents with depression had a significantly higher chance of being smokers compared to respondents without depression (OR = 1.347. 95% CI: 1.042–1.741). Further, respondents with anxiety had a significantly higher chance of being smokers compared to respondents without anxiety (OR = 1.401. 95% CI: 1.095–1.793). This finding showed that respondents with stress had a higher chance of being smokers compared to respondents without stress (OR = 1.211. 95% CI: 0.939–1.560), but it was not statistically significant at a 0.05 level.

## 4. Discussion

### 4.1. Prevalence of Smoking among Adult Men and Women

In Malaysia, only a few studies have examined the prevalence of smoking in the general population, with most studies focusing on adolescents. Among those studies is the study published in 2014 by Al-Naggar et al. [22] assessing prevalence of smoking in Malaysia among the general public with data collection completed between June 2012 to September 2012. Both the Al-Naggar et al. (2014) study and this study have a few similarities, namely that data collection was by face to face interview and the time period was in 2012. Both Shah Alam and Kerinchi have predominantly Malay inhabitants, and both locations are urban areas, with Kerinchi being of low socioeconomic status, while some areas of Shah Alam are affluent. The difference between these two studies was the location of the collection of data. The Al-Naggar et al. (2014) [22] study collected data at a restaurant while our study collected data at home. The prevalence of tobacco use in our study was only 17.64%, and this is much lower than the study by Al-Naggar et al. (2014) [22] where the prevalence of smoking was 57%. This higher value may be due to the respondents being at a restaurant, and may not have the chance to conceal that they are smokers when approached by the interviewers. In addition, if respondents are sitting with their friends who are also smokers, they may be more likely to be open compared to our study where the respondents were approached at home and may have to conceal smoking from their families. Hence, it is proposed that location of the assessment may be important. In 2011, the Global Adult Tobacco Survey (GATS) [23] estimated that approximately 23% of Malaysia’s population smoked (approximately 4,844,800 persons). However, this value is closer to our result of 17.64% and is in line with the apparent decreasing use of tobacco.

In our study, 35.5% of males and 1.83% of females are smokers. In a study by Lim et al. (2013), 15,639 participants elicited a smoking prevalence of 46.5% among males, which is much higher than our study [7]. This is comparable to the study by Rampal et al. (2006) [24] and the Malaysian Non Communicable Disease Surveillance-1 in 2011 [25] which found smoking rates among males in Malaysia to be 47.2% and 40.9% respectively. These values are similar to the rates from the Global Adult Tobacco Survey (GATS) in 2011 which reported that 43.6% of Malaysian males aged 15 years and above were current smokers [23]. The face to face interview of our study may have contributed to under reporting. In addition, there may be an increased awareness by people of the ill effects of nicotine and the carcinogenic nature of cigarettes due to social media leading to decreased use. Males were more likely to use cigarettes compared to females. This has been a consistent finding. In our study, age and level of education were not significantly correlated with smoking. Additionally, housewives and unemployed persons were less likely to smoke compared to civil servants. In a previous study conducted on 112 people working at two municipalities in Malaysia, 37.56% of civil servants were found to have nicotine dependence [26].

### 4.2. Sociodemographic Factors and Patterns of Use

Generally, it was a Malay male in the age group of 18 to 29 who was a smoker, despite age not being statistically significant. This finding has been replicated in other studies [7]. In our study, 75.8% of smokers were introduced to the first cigarette through friends. The age of initiation into smoking in Malaysia was 18.3 years [7], with 82.3% of the respondents smoking daily and 47.9 % trying to quit in the last 3 months. According to GATS (2011) [23], 48.6% of smokers made attempts to quit in the preceding 12 months. Better interventions need to be put in place to assist quitting [7].

### 4.3. Associations between Stress, Anxiety and Depression in Smoking

An association between smoking, and depression and anxiety was found. Another study reported that feelings of sadness and loneliness were cues to smoke [26]. Major depression has been associated in the past with increased rates of daily smoking and elevated rates of nicotine dependence [27]. Anxiety has also shown to be statistically higher in smokers than non-smokers [28]. Conversely, cigarette smoking and nicotine dependence appeared to be risk factors for the development of some anxiety disorders, such as panic disorder or generalized anxiety disorder. This was also true for major depression [29]. Being depressed also increased the risk of the smoker to use cigarettes daily [29].

### 4.4. Limitations

Location of the assessment was at home and respondents may conceal smoking. Using face to face interviews may yield lower smoking prevalence.

### 4.5. Conclusion

The location of the assessment and not using face to face interviews may be an important factor in determining smoking prevalence. Most Malaysian smokers tend to be either young Malay males or civil servants, especially those who are divorced. More comprehensive anti-smoking policy measures are needed in order to assist those intending to quit. Depression and anxiety, but not stress, were significantly associated with smokers.

## 5. Conclusions

This study demonstrates that the chance of depression and anxiety is prevalent, especially amongst Malay men who smoke and who are also from a low socioeconomic class. Therefore, it is recommended that screening for depression and anxiety should be routinely performed in the primary care setting and in the population based health screening to intervene early in patients who smoke.

## Figures and Tables

**Table 1 ijerph-16-01762-t001:** Distribution of demographic variables.

Sociodemographic Characteristics	Non-Smoker *n* (%)	Smoker *n* (%)	*p*-Value
Age			
18 to 29	547 (84.3%)	102 (15.7%)	0.427
30 to 39	326 (80.7%)	78 (19.3)	
40 to 49	349 (81.2%)	81 (18.8%)	
50 to 59	254 (80.9(%)	60 (19.1%)	
60 and above	162 (84.4%)	30 (15.6%)	
Gender			
Male	603 (64.5%)	332 (35.5%)	<0.001
Female	1035 (98.2%)	19 (1.8%)	
Ethnicity			
Malay	1304 (81.1%)	304 (18.9%)	0.037
Chinese	18 (85.7%)	3 (14.3%)	
Indian	298 (87.9%)	41 (12.1%)	
Other Bumiputra	5 (100%)	0 (%)	
Others	13 (81.3%)	3 (18.8%)	
Religion			
Islam	1322 (81.2%)	307 (18.8%)	0.042
Christian	19 (82.6%)	4 (17.4%)	
Buddhist	18 (85.7%)	3 (14.3%)	
Hindu	276 (88.5%)	36 (11.5%)	
Sikh	3 (75.0%)	1 (25.0%)	
Education Level			
No formal Education	127 (85.2%)	22 (14.8%)	0.073
Primary School	194 (81.9%)	43 (18.1%)	
Secondary School	978 (80.8%)	232 (19.2%)	
Tertiary	339 (86.3%)	54 (13.7%)	
Marital Status			
Married	966 (79.2%)	254 (20.8%)	<0.001
Divorced	92 (94.8%)	5 (5.2%)	
Widow	74 (96.1%)	3 (3.9%)	
Single	506 (85.0%)	89 (15%)	
Status Occupation			
Civil Servant	105 (76.6%)	32 (23.4%)	<0.001
Private Sector	700 (79.0%)	186 (21.0%)	
Self-employed	135 (69.2%)	60 (30.8%)	
Student	134 (94.4%)	8 (5.6%)	
Housewife	158 (77.1%)	47 (22.9%)	
Unemployed	360 (98.9 %)	4 (1.1%)	
Retiree	46 (76.7%)	14 (23.3%)	

**Table 2 ijerph-16-01762-t002:** Patterns of smoking among respondents.

Variable	Level	Frequency	Percent
Route of administration	Oral	18	5.1
	Smoking	316	90
	N/A	17	4.8
Who introduced you to smoking?	Friends	266	75.8
	Other family member	3	0.9
	Acquaintance	19	5.4
	NA	63	17.7
How frequent is the usage	Once or twice ever	27	7.7
	Once a month	4	1.1
	Once a week	6	1.7
	Everyday or almost every day	289	82.3
	N/A	25	7.2
How frequently the usage brought problems	Never	232	66.1
	Once or twice	47	13.2
	A month	11	3.1
	A week	15	4.3
	Everyday or almost every day	46	13.1
How frequently the smoking makes you fail performing daily routine	Never	303	86.3
	Once or twice	27	7.7
	A month	4	1.1
	A week	2	0.6
	Everyday or almost every day	15	3.7
Anybody concerned about this usage?	Never	73	20.8
	Yes, but not in the past 3 months	88	25.1
	Yes, in the past 3 months	190	54.1
Tried to quit but couldn’t control	Never	106	30.2
	Yes, but not in the past 3 months	77	21.9
	Yes, in the past 3 months	168	47.9

**Table 3 ijerph-16-01762-t003:** Results of stepwise multivariate logistic regression between sociodemographic characteristics and smoking.

Sociodemographic Characteristics	*p*-Value	OR	95% (CI)	
			Lower	Upper
Gender				
Male		Reference		
Female	0.0001	0.039	0.023	0.066
Ethnicity				
Malay		Reference		
Chinese	0.804	1.302	0.162	10.490
India	0.751	0.727	0.101	5.215
Other Bumiputra	0.999	0.000	0.000	
Others	0.696	1.396	0.261	7.451
Education level				
No formal Education		Reference		
Primary School	0.374	0.733	0.370	1.454
Secondary School	0.159	0.636	0.339	1.194
Tertiary	0.080	0.530	0.260	1.079
Marital Status				
Married		Reference		
Divorced	0.001	1.999	1.310	3.050
Widow	0.635	1.317	0.423	4.098
Single	0.374	1.926	0.454	8.164
Occupational status				
Civil Servant		Reference		
Private Sector	0.293	0.766	0.466	1.259
Self-employed	0.559	1.190	0.664	2.132
Student	0.338	0.666	0.290	1.529
Housewife	0.002	0.242	0.097	0.606
Unemployed	0.012	0.228	0.072	0.723
Retire	0.270	1.428	0.758	2.689

**Table 4 ijerph-16-01762-t004:** Relationship between Depression Anxiety Stress Scale and smoking.

Psychological Problems	Smoker (%)	Non-Smoker (%)	*p*-Value	Crude OR (95% CI)	*p*-Value	Adjusted OR (95% CI)
No Depression	1252 (83.5%)	248 (16.5%)	reference			
Depression	386 (78.9%)	103 (21.1%)	0.123	1.308 (0.944–1.622)	0.014	1.347 (1.042–1.741)
No Anxiety	1199 (83.8%)	232 (16.2%)	reference			
Anxiety	439 (78.7%)	119 (21.3%)	0.047	1.308 (1.003–1.705)	0.005	1.401 (1.095–1.793)
No Stress	1211 (83.1%)	246 (16.9%)	reference			
Stress	427 (80.3%)	105 (19.7%)	0.699	1.055 (0.804–1.385)	0.080	1.211 (0.939–1.56)

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
