# Peer review of "The Prevalence of Smoking, Determinants and Chance of Psychological Problems among Smokers in an Urban Community Housing Project in Malaysia"

_ijerph, 2019, doi:10.3390/ijerph16101762_

Round 1
Reviewer 1 Report
The researchers of this study assessed the prevalence, patterns of smoking and socio demographic factors among residents of 4 community housing projects in Kerinchi. The article is very interesting, but does have some limitations:
The authors state that they want to assess the prevalence, patterns of smoking and socio demographic factors among Kerinechi residents, but the study is carried out in 4 community housing projects. Unless these projects are a good representation of Kerinechi residents, I don’t think their methodology is adequate.
It would be important to state which scales they used for Nicotine Abuse, and the exact cut offs for poverty.
The authors mention the assumptions they used to conduct their sample size calculations; it would be nice to know where they got these assumptions.
The article might benefit from a
The data of this study is 6.5 years old, not sure how relevant the findings are of today’s situation.
Author Response
Reviewer 1
The researchers of this study assessed the prevalence, patterns of smoking and socio demographic factors among residents of 4 community housing projects in Kerinchi. The article is very interesting, but does have some limitations:
1) The authors state that they want to assess the prevalence, patterns of smoking and socio demographic factors among Kerinechi residents, but the study is carried out in 4 community housing projects. Unless these projects are a good representation of Kerinechi residents, I don’t think their methodology is adequate.
Reply: The original aim was to assess the health status of the low-income community in Kuala Lumpur. The nearby Lembah Pantai area was selected mainly due to logistic reasons after discussion with Dewan Bandaraya Kuala Lumpur that is the local municipality. The aim of this study was to eventually take care of the low-income people nearby the University Malaya. There are six PPR in Lembah Pantai area and four PPR were selected randomly.
2) It would be important to state which scales they used for Nicotine Abuse, and the exact cut offs for poverty.
Reply: The ASSIST scale (Alcohol, Smoking and Substance Involvement Screening Test) was used to assess nicotine abuse.
Reference
WHO ASSIST Working Group (2002). The Alcohol, Smoking and Substance Involvement Screening Test (ASSIST): development, reliability and feasibility. Addiction, 97 (9): 1183-1194.
Humeniuk RE, Ali RA, Babor TF, Farrell M, Formigoni ML, Jittiwutikarn J, Boerngen de Larcerda R, Ling W, Marsden J, Monteiro M, Nhiwhatiwa S, Pal H, Poznyak V & Simon S (2008). Validation of the Alcohol Smoking and Substance Involvement Screening Test (ASSIST). Addiction 103(6): 1039-1047
· The poverty line in Malaysia is RM 800. The households that earn Rm 800 and below are considered to be in the poverty line.
However for our study, the criteria to be included was that the household does not have an income above RM 2000. Hence some household were not below the poverty line.
3) The authors mention the assumptions they used to conduct their sample size calculations; it would be nice to know where they got these assumptions.
Reply: According to Prof Tin Su the assumptions are merely just assumptions. There are no references for it. Prevalence of 50% psychiatric illness was used as that would yield the largest sample size for this study.
4) The data of this study is 6.5 years old, not sure how relevant the findings are of today’s situation. Despite the data is 6.5 years old but the authors felt that the findings are still relevant
Reply: This study is important because
· There are studies published after 2012 that have yielded similar results as our study, that is smokers have higher chance to be depressed or anxious as compared to the general population. However these studies were not conducted in Malaysia.
· Our study focusses on a low socioeconomic area and as we know social determinants act differently across different income groups and geographical locations.
· This study is important as it may deepen the understanding and contribute information towards nicotine use especially in low socioeconomic population in Malaysia and may provide insight into how to prevent or mitigate nicotine use amongst the low socioeconomic population.
Given below are four other studies published in recent years (2017 -2019) that show various associations between nicotine use and psychological problems namely stress, depression and anxiety
1. Rocha SAV, Hoepers ATC, Fröde TS, Steidle LJM, Pizzichini E, Pizzichini MMM Prevalence of smoking and reasons for continuing to smoke: a population-based study. J Bras Pneumol. 2019 Mar 28;45(4):e20170080. doi: 10.1590/1806-3713/e20170080
2. Secades-Villa R, Weidberg S, González-Roz A, Reed DD, Fernández-Hermida JR
Cigarette demand among smokers with elevated depressive symptoms: an experimental comparison with low depressive symptoms. Psychopharmacology (Berl). 2018 Mar;235(3):719-728. doi: 10.1007/s00213-017-4788-1. Epub 2017 Nov 15.
3. Lise S. Skov-Ettrup, Børge G. Nordestgaard, Christina B. Petersen, Janne S. Tolstrup, Does High Tobacco Consumption Cause Psychological Distress? A Mendelian Randomization Study Nicotine & Tobacco Research, Volume 19, Issue 1, 1 January 2017, Pages 32–38.
4. Yunwei Yan, Shuxia Sun, Songyuan Deng, Jicheng Jiang, Fujiao Duan, Chunhua Song
A systematic review of anxiety across smoking stages in adolescents and young adults.
Subst Use Misuse. 2019 Apr 3:1-8.
Although this study addresses a highly relevant issue in public health, the design, data analysis and reporting lack consistency and rigor. As it is, its scientific contribution is average. Overall, extensive editing of language is required.
Reviewer 2 Report
Although this study addresses a highly relevant issue in public health, the design, data analysis and reporting lack consistency and rigor. As it is, its scientific contribution is average. Overall, extensive editing of language is required.
Title:
- After reading all the manuscript, it is not clear if the title is describing adequately its content. Is the objective of the paper to describe prevalence of smoking in this community or to assess the prevalence of “psychological problems” among smokers? (Or both)?;
- The term “psychological problems” raises doubts from the psychological literature standpoint. It has a negative tone and is vague. Maybe one suggestion could be “psychopathological symptoms”?
Abstract:
- It is not clear if the first objective is descriptive? It should be also clarified the aim: “to assess the prevalence” – of what?;
- The objective could be formulated in one sentence;
- The reporting of the results is not consistent: some have p-values, others do not;
- It is not clear why the conclusion only focus the use of the DASS scale; it is not coherent with the objectives of the study presented.
Introduction:
- Overall, the Introduction is written in a very simplistic way, not taking into account the huge research done in the field of smoking determinants, namely social determinants, and all the nuances that should be taken into account (e.g., page 1, line 39, “This is due to the fact that tobacco brings euphoric effects”. There are other literature reporting different physical and emotional consequences of smoking for individuals that should be considered);
- Page 2, lines 44-45 – the sentence about the prevalence of smoking in the USA, as it is, is decontextualized and referenced by very old references (e.g., the reports from the USDHHS);
- The use of the expression “exact number” should be replaced in the objective, taking into account the authors cannot guarantee they are able to gather that data;
- The hypothesis is not clear enough: neither the variables are identified nor the direction in which it is expected that some sociodemographic factors are related with “prevalence”. It is also needed to clearly state to which this prevalence refers to (prevalence of smoking, prevalence of psychopathological symptoms?).
Methodology:
- The assumption that 50% of the population has one psychiatric morbidity should be referenced, if possible;
- Regarding the DASS Scale, is it usually applied by a researcher or it could be self-administered? The possible bias resulting from its application by a researcher should be also included and discussed in the limitations of the study;
- The section Statistical analyses is almost inexistent. Although the choices for the statistical tests appear along the text, they should be presented in this section with detail.
Results:
- The results could be described before the respective table;
- Table 1. Several formatting problems exist in the tables. E.g.: one line with “n (%)” is missing; the sociodemographic characteristics may not be considered as “risk factors”; in the variable age, is the last category “>60” including those who have 60 years? In that case, the symbol should be replaced to ≥;
- Table 2. It is not clear what is meant by the question “how frequently the usage brought problems?”. What kind of problems are the researchers referring to? Information about the total number of individuals in each table is missing;
- Table 3. It is not clear why the authors did not present only one table with the results from Tables 1 and 3 (presenting the results from the logistic regression – crude and adjusted odds ratios). Presenting the ORs and the 95% CI, the presentation of the p-value can be omitted, taking into account it is redundant.
- Also, in Table 3, the authors did not describe in the table (as footnote) for which variables is the model adjusted?
- The description of the results in this section is vague (e.g., page 6, lines 132-135) and do not allow to understand the direction of the results, which only appears further in the text.
Conclusion:
- Line 210 - Conclusions drawn from the study are not completely sustained by data provided. How can the authors state that the trend of smoking appears to be in a down trend using a cross-sectional design?
- Lines 217 – 221 - The Conclusion in the final section of the paper only highlight results regarding the instrument, which, again, is not clearly presented as the main objective of the paper. Also, the data collected does not allow to draw conclusions about the content/construct validation of the instrument based on this application. What is the guarantee that the instrument is assessing depression and anxiety symptoms?
Author Response
Dear Reviewer
Kindly find my response.
Thank you for reading and providing suggestions.
Reviewer 2
Title:
1) After reading all the manuscript, it is not clear if the title is describing adequately its content. Is the objective of the paper to describe prevalence of smoking in this community or to assess the prevalence of “psychological problems” among smokers? (Or both)?
Reply: This paper is describing the prevalence of smoking in the Kerinchi community and not the prevalence of psychological problems in the Kerinchi community.
The authors agree with the reviewer and hence have changed the title from “The Prevalence, Determinants and Chance of Psychological Problems among Smokers in Urban Community Housing Project in Malaysia” to “The Prevalence of Smoking, Determinants and Chance of Psychological Problems among Smokers in Urban Community Housing Project in Malaysia.”
2) The term “psychological problems” raises doubts from the psychological literature standpoint. It has a negative tone and is vague. Maybe one suggestion could be “psychopathological symptoms”?
Reply: The term “psychological problems” is frequently used by psychiatrists to describe psychological morbidity.
Abstract:
1) It is not clear if the first objective is descriptive? It should be also clarified the aim: “to assess the prevalence” – of what?
Reply: The prevalence studied is of prevalence of smoking in the Kerinchi community and not prevalence of psychological problems in Kerinchi community.
2) The objective could be formulated in one sentence;
Reply: The authors agree with the reviewer and will formulate the objective in one sentence. The objective has been changed from “To assess prevalence, pattern of smoking and sociodemographic factors among Kerinchi residents in Kuala Lumpur. To identify the association between smoking, stress, anxiety and depression.” to “To assess prevalence, pattern of smoking, sociodemographic factors among Kerinchi residents in Kuala Lumpur as well as to identify the association between smoking, stress, anxiety and depression.
3) The reporting of the results is not consistent: some have p-values, others do not;
Reply: Only table 2 has no p value and it is just descriptive results among smokers that’s why no p value was calculated
4) It is not clear why the conclusion only focus the use of the DASS scale; it is not coherent with the objectives of the study presented.
Reply: The authors agree with the reviewer and hence the conclusion has been changed from “This study demonstrates that DASS is a useful screening tool to assess chance of developing depression and anxiety among smokers, especially Malay men from low socio economic class. Therefore we recommend that DASS should be routinely used in the primary care setting and in the population based health.” to “This study demonstrates that depression and anxiety is prevalent especially amongst Malay men who smoke and who are also from low socio economic class. Therefore we recommend that screening for depression and anxiety should be routinely performed in the primary care setting and in the population based health screening to intervene early in patients who smoke.”
Introduction:
1) Overall, the Introduction is written in a very simplistic way, not taking into account the huge research done in the field of smoking determinants, namely social determinants, and all the nuances that should be taken into account (e.g., page 1, line 39, “This is due to the fact that tobacco brings euphoric effects”. There are other literature reporting different physical and emotional consequences of smoking for individuals that should be considered);
Reply: The authors agree with the reviewer that there is much research done on determinants of smoking especially social determinants. The authors also agree that nicotine has emotional and physical effects and have hence amended the introduction.
Line 39
The authors have changed the sentence from “This is due to the fact that tobacco brings euphoric effects [1].” to “This is due to the fact that tobacco has been shown to have antidepressant effects [5].”
Line 46
The sentence has been inserted
“Social determinants have been shown to play a large role as smoking is not equal amongst the various social determinants and is prevalent amongst people living in poverty [8].”
Line 55
The following sentence has been inserted.
“80% of smokers come from low and middle income countries[8].”
Line 57
The authors agree with the reviewer that social determinants play a big role in nicotine use and have thus changes the sentence “A low level of education, makes one susceptible to smoking [13]” to “A low level of education and low income, makes one susceptible to smoking [13,14]”.
2)Page 2, lines 44-45 – the sentence about the prevalence of smoking in the USA, as it is, is decontextualized and referenced by very old references (e.g., the reports from the USDHHS);
Reply: The authors agree with the views of the reviewer and have thus removed the following sentence. “Prevalence of smoking in USA was 25% in 1993 then decreased to 23.3% in 2000[8].”
3)The use of the expression “exact number” should be replaced in the objective, taking into account the authors cannot guarantee they are able to gather that data;
Reply: The authors agree with the reviewer and hence the word “exact” has been removed from the sentence
4) The hypothesis is not clear enough: neither the variables are identified nor the direction in which it is expected that some sociodemographic factors are related with “prevalence”. It is also needed to clearly state to which this prevalence refers to (prevalence of smoking, prevalence of psychopathological symptoms?).
Reply: The hypothesis was “There is an association between sociodemographic factors and prevalence among Kerinchi residents in Kuala Lumpur.” This was changed to “There is an association between prevalence of smoking, socioeconomic factors, depression and anxiety amongst smokers in Kerinchi residents in Kuala Lumpur”
Methodology:
1) The assumption that 50% of the population has one psychiatric morbidity should be referenced, if possible;
Reply: According to Prof Tin Su the assumptions are merely just assumptions. There are no references for it. Prevalence of 50% psychiatric illness was used as that would yield the largest sample size for this study.
2) Regarding the DASS Scale, is it usually applied by a researcher or it could be self-administered? The possible bias resulting from its application by a researcher should be also included and discussed in the limitations of the study;
Reply: The DASS scale was administered by 10 trained data collectors who were not the researchers. Hence there is no bias and was not discussed in the limitation section.
3) The section Statistical analyses is almost inexistent. Although the choices for the statistical tests appear along the text, they should be presented in this section with detail. ;
Reply: All statistical analyses were performed using SPSS version 23. Descriptive statistics such as frequency and percentage was applied for categorical data. Inferential statistics including Chi square test and univariate and stepwise multivariate logistic regression were carried out to study the relationship between variables and outcome (smoker vs non-smoker). A p-value of ≤ 0.05 was considered statistically significant.
Results:
-The results could be described before the respective table;
1)Table 1. Several formatting problems exist in the tables. E.g.: one line with “n (%)” is missing; the sociodemographic characteristics may not be considered as “risk factors”; in the variable age, is the last category “>60” including those who have 60 years? In that case, the symbol should be replaced to ≥;
Reply: n(%) has been added
The symbol ≥ has been rectified
2) Table 2. It is not clear what is meant by the question “how frequently the usage brought problems?” What kind of problems are the researchers referring to? Information about the total number of individuals in each table is missing;
Reply: The researchers are referring to any sort of problem that was caused by nicotine use.
3)Table 3 It is not clear why the authors did not present only one table with the results from Tables 1 and 3 (presenting the results from the logistic regression – crude and adjusted odds ratios). Presenting the ORs and the 95% CI, the presentation of the p-value can be omitted, taking into account it is redundant.
Reply: Table 1 consists of univariate analysis between sociodemographic variables and smoking while table 3 consists of the results of stepwise multivariate logistic regression and only significant variables were entered to this model
4) Also, in Table 3, the authors did not describe in the table (as footnote) for which variables is the model adjusted?
Reply: This table consists of the results of multivariate logistic regression and that is why we don’t have crude OR. Only Adjusted OR based on multivariate methods was reported. . The title of the table was changed from “Relationship between socio demographic characteristics and smoking” to “Results of stepwise multivariate logistic regression between socio demographic characteristics and smoking.”
5) The description of the results in this section is vague (e.g., page 6, lines 132-135) and do not allow to understand the direction of the results, which only appears further in the text.
Reply: This section has been corrected.
Conclusion:
1) Line 210 - Conclusions drawn from the study are not completely sustained by data provided. How can the authors state that the trend of smoking appears to be in a down trend using a cross-sectional design?
Reply: Line 210
The authors agree with the reviewer that the trend of smoking cannot be inferred from a cross sectional study and have hence removed the sentence” The prevalence of smoking among Malaysian males appears to be on a down trend.” from the conclusion.
2) Lines 217 – 221 - The Conclusion in the final section of the paper only highlight results regarding the instrument, which, again, is not clearly presented as the main objective of the paper. Also, the data collected does not allow to draw conclusions about the content/construct validation of the instrument based on this application. What is the guarantee that the instrument is assessing depression and anxiety symptoms?
DASS is a well known valid and reliable scale to screen for anxiety and depression
Reply: Lines 217 – 221
The authors agree with the reviewer that the function of the DASS instrument in assessing stress, depression ad anxiety was not part of the main objective of the paper and as such have not mentioned it in the conclusion.
The conclusion has been changed from “This study demonstrates that DASS is a useful screening tool to assess chance of developing depression and anxiety among smokers, especially Malay men from low socioeconomic class. Therefore we recommend that DASS should be routinely used in the primary care setting and in the population based health screening to intervene early those patients who may have depression and anxiety.” to” This study demonstrates that chance of depression and anxiety is prevalent especially amongst Malay men from low socio economic class who smoke. Therefore we recommend that screening for depression and anxiety should be routinely performed in the primary care setting and in the population based health screening to intervene early in patients who smoke.”
Round 2
Reviewer 2 Report
The authors improved the text so that it could be published.
Author Response
Thank you very much for your comments.